# Evaluating the reactivation of herpesviruses and inflammation as cardiovascular and cerebrovascular risk factors in antiretroviral therapy initiators in an African HIV-infected population (RHICCA): a protocol for a longitudinal cohort study

Ingrid Peterson,[1,2] Ntobeko Ntusi,[3] Kondwani Jambo,[1,2] Christine Kelly,[2,4] Jacqueline Huwa,[2] Louise Afran,[2] Joseph Kamtchum Tatuene [ID] ,[2,5] Sarah Pett,[6,7] Marc Yves Romain Henrion,[1,2] Joep Van Oosterhout,[8,9] Robert S Heyderman,[2,10] Henry Mwandumba,[1,2] Laura A Benjamin [ID] ,[11,12] On behalf of the Investigators of the RHICCA study

For numbered affiliations see end of article.

**Correspondence to**
Dr Laura A Benjamin;
l.benjamin@liverpool.ac.uk

## ABSTRACT

**Introduction** In Sub-Saharan Africa, the rising rates of cerebrovascular and cardiovascular diseases (CBD/CVD) are intersecting with an ageing HIV-infected population. The widespread use of antiretroviral therapy (ART) may confer an additive risk and may not completely suppress the risk associated with HIV infection. High-quality prospective studies are needed to determine if HIV-infected patients in Africa are at increased risk of CBD/CVD and to identify factors associated with this risk. This study will test the hypothesis that immune activation and dysfunction, driven by HIV and reactivation of latent herpesvirus infections, lead to increased CBD/CVD risk in Malawian adults aged ≥35 years.

**Methods and analysis** We will conduct a single-centre, 36-month, prospective cohort study in 800 HIV-infected patients initiating ART and 190 HIV-uninfected controls in Blantyre, Malawi. Patients and controls will be recruited from government ART clinics and the community, respectively, and will be frequency-matched by 5-year age band and sex. At baseline and follow-up visits, we will measure carotid intima-media thickness and pulse wave velocity as surrogate markers of vasculopathy, and will be used to estimate CBD/CVD risk. Our primary exposures of interest are cytomegalovirus and varicella zoster reactivation, changes in HIV plasma viral load, and markers of systemic inflammation and endothelial function. Multivariable regression models will be developed to assess the study's primary hypothesis. The occurrence of clinical CBD/CVD will be assessed as secondary study endpoints.

**Ethics and dissemination** The University of Malawi College of Medicine and Liverpool School of Tropical Medicine research ethics committees approved this work. Our goal is to understand the pathogenesis of CBD/CVD among HIV cohorts on ART, in Sub-Saharan Africa, and provide data to inform future interventional clinical trials. This study runs between May 2017 and August 2020. Results of the main trial will be submitted for publication in a peer-reviewed journal.

<div style="background:#c8cce8">

### Strengths and limitations of this study

- ▶ This is one of the first large-scale studies in Sub-Saharan Africa to explore the relationship between HIV infection, latent herpesviruses, inflammation, and cardiovascular and cerebrovascular diseases, immediately after starting antiretroviral therapy (ART).
- ▶ Clinical events and death will be comprehensively reviewed through an endpoint review committee using strict diagnostic criteria for events based on those used in the INSIGHT (International Network for Strategic Initiatives in Global HIV Trials) network or validated verbal autopsy for death with limited data.
- ▶ Because of the recent roll-out of ART in asymptomatic patients, there will be an absence of ART-naïve population, limiting our ability to explore the impact of ART.
- ▶ Approximately one-third of strokes will be asymptomatic and we anticipate not capturing some of these; however, multiple cerebral infarcts without a focal neurological deficit will manifest as cognitive impairment, which we will screen for and corroborate with MRI in a small number of symptomatic cases.
- ▶ Two-thirds of myocardial infarction will be silent and could potentially be missed, and in a nested group we will use digital ECG to evaluate this further.

</div>

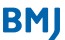

**Trial registration number** ISRCTN42862937.

## INTRODUCTION

The growing epidemic of cerebrovascular disease (CBD; eg, stroke) and cardiovascular disease (CVD; eg, myocardial infarction (MI)) now intersects with the HIV epidemic.[1] Countries such as Malawi have an adult HIV prevalence of approximately 10%.[2] There is an increased life expectancy among people living with HIV, largely because of the successful scale-up of antiretroviral therapy (ART).[3] In Europe and the USA, HIV is associated with a 50% increased risk of CVD compared with HIV-uninfected populations,[4] attributable to long-term ART use and HIV per se.[4 5] HIV infection is also associated with a 1.8-fold increased risk of all-cause heart failure in US veterans.[6] Our recent case–control study of stroke in Malawian adults is one of several examples that demonstrate a high risk of HIV infection associated with stroke and heart disease, pointing to a considerable and unappreciated CBD/CVD risk among patients with HIV in this setting.[7–10]

There are reports of geographical differences in the distribution of CVD risk factors, supporting the argument that evidence derived from high-income countries cannot be applied to Sub-Saharan (SSA).[11] Addressing this knowledge gap is essential to the development of clinical drug trials for primary prevention of CBD/CVD among individuals living with HIV. Vasculopathy due to accelerated atherosclerosis, arterial stiffening and vasculitis are the major mechanisms believed to underlie the CBD/CVD burden.[12 13] It is hypothesised that despite viral suppression,

low-grade HIV virus replication and the associated host systemic inflammation are important drivers of this vasculopathy (figure 1). In patients receiving ART, HIV antigenaemia, partly resulting from HIV persistence in sanctuary sites, incomplete virological suppression and virological resurgence, drives the chronic immune activation observed in about 20% of ART patients in SSA.[14] This immune state is characterised by ongoing activation and senescence of cell-mediated immunity,[15 16] increased monocyte/macrophage activation, stimulation of the interleukin-6 (IL-6) pathway and production of acute phase proteins.[17–19] Activation of the IL-6 pathway is established with atherosclerosis[20 21] and may also contribute to non-atherosclerotic vasculopathy. Inflammation alone confers a two-fold increased risk of clinical CBD/CVD events.[22] The current push to introduce more effective ART regimens and to start treatment soon after HIV diagnosis is made may reduce inflammation and in turn CBD/CVD risk.[23] However, there is growing evidence of chronic inflammation in HIV despite achieving the goal of therapy, which is long-term suppression (<50 copies/mL) of plasma viral load, suggesting adjunctive therapy may be required.[24–26]

In addition to HIV, there is compelling evidence that reactivation of latent herpesviruses may be an important cause of vasculopathy. In HIV-uninfected elderly populations from high-income settings, latent cytomegalovirus (CMV) infection drives dysregulation of cell-mediated immunity,[15 27–29] not dissimilar to what is described in HIV-associated immune activation.[29] CMV and other viral proteins have been found in atherosclerotic plaques.[20]

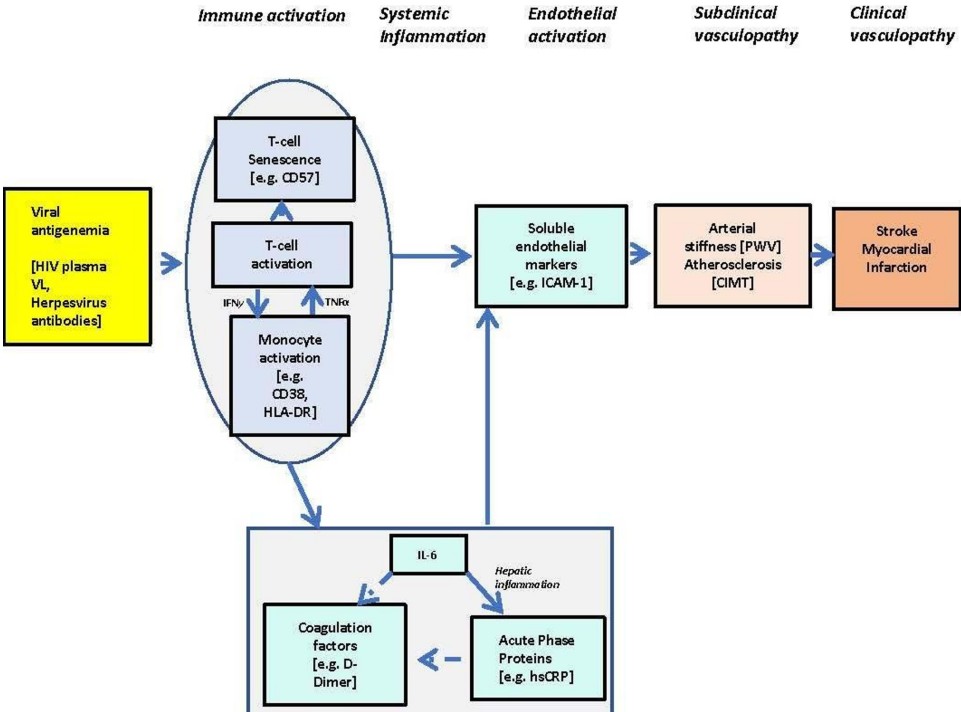

**Figure 1** Hypothetical pathway of the interplay between chronic viruses, immune activation, systemic inflammation, endothelial activation and vasculopathy. CIMT, carotid intima-media thickness; IL-6, interleukin-6; PWV, pulse wave velocity; VL, Viral Load; HLA-DR, Human Leukocyte Antigen-DR isotype; IFNγ, interferon gamma; TFNα, Tumour Necrosis Factor alpha; ICAM-1, intercellular adhesion molecules-1 and hsCRP, high-sensitivity C-reactive protein.

Varicella zoster virus (VZV) can directly infect the vascular endothelium to cause vasculitis and subsequent stroke and was found to be the most common opportunistic infection (prevalence 15%) in a study of HIV-infected patients with stroke in Malawi.[12] The seroprevalence of herpesviruses is high in SSA,[30] particularly in HIV-infected populations.[16]

The involvement of herpesviruses in the mechanistic pathway for CBD/CVD is compelling and may offer additional therapeutic avenues, especially for CMV and VZV. However, our understanding is incomplete, and its population impact is yet to be defined. It is important to determine if, in addition to ART, there is a role for other pharmacological interventions targeting latent viral infections or downstream inflammatory pathways to reduce vasculopathy in HIV-infected patients on ART. Previous work from North America supports the potential of treating reactivated herpesviruses.[31] Furthermore, there are opportunities for intervention using the recently licensed letermovir, a treatment for CMV. By focusing on HIV and herpes viral antigenaemia and immune dysregulation as mechanisms of vasculopathy, this study will identify subgroups of HIV-infected patients on ART at high risk of CBD/CVD, the timing of CBD/CVD risk in such patients, as well as potential targets for intervention. Ultimately, the study goal is to investigate whether reactivation of herpesviruses and inflammation are cardiovascular and cerebrovascular risk factors in antiretroviral therapy initiators in an African HIV-infected population (RHICCA).

### Study objectives

This study will test the hypothesis that immune activation and dysfunction, driven by HIV and reactivation of latent CMV/VZV herpesvirus infections, lead to increased CBD/CVD risk in adults aged ≥35 years in SSA. We will address this through the following objectives:

1. To determine if progression of the surrogate marker of CBD/CVD or occurrence of new-onset vasculopathy is higher in adults aged ≥35 years with HIV infection on ART compared with those without HIV.
2. To determine if progression of surrogate markers of CBD/CVD or occurrence of new-onset vasculopathy is higher in adults aged ≥35 years with HIV/herpes viral antigenaemia or chronic immune activation compared with those without HIV/herpes viral antigenaemia or chronic immune activation. Specifically, we will determine if progression of surrogate markers or new-onset vasculopathy is higher in the following:
   a. In ART patients with reactivated latent herpes viral infection, compared with those without reactivated latent herpes viral infection.
   b. In ART patients with the highest 25% of markers for immune activation, inflammation or endothelial activation compared with the bottom 25%.
   c. In ART patients with incomplete virological suppression or virological resurgence of HIV, compared with those with suppressed HIV plasma viral load.

The secondary study objectives are to determine if viral antigenaemia or chronic immune activation increases the occurrence of the following clinical events: (1) stroke, (2) MI, (3) angina (excluding MI), (4) peripheral vascular disease (PVD), (5) all-cause death/vascular-related death and (6) immune reconstitution inflammatory vasculopathy.

## METHODS AND ANALYSIS
### Study design

To address objective 1, we will conduct a single-centre, 36-month, prospective cohort study in 800 HIV-infected patients initiating ART and 190 HIV-uninfected adults aged ≥35 years. HIV-infected and HIV-uninfected participants will be frequency-matched by 5-year age band and sex. On a 6-monthly basis, we will measure markers of viral infection, inflammation and endothelial function along with surrogate markers for CBD/CVD (table 1).

### Study setting

This study will recruit consecutive ART patients from the ART clinic of Queen Elizabeth Central Hospital (QECH) and ART clinics in several Blantyre City community health centres (CHCs). These clinics collectively initiate over 100 HIV-infected patients aged ≥35 years onto ART each month. HIV-uninfected adults will be selected from pre-ART counselling sessions, and from randomly selected households in the community by two-stage random sampling (of households and individuals within households) from a previously enumerated sampling frame in the CHC catchment areas.[32] All study procedures will be conducted at the QECH, which is located adjacent to the Malawi-Liverpool-Wellcome Trust Clinical Research Programme (MLW). The QECH also hosts a 0.35T MRI facility, which will contribute to characterising our secondary endpoints.

### Study participants

Study inclusion criteria will be (1) age ≥35 years and (2) resident of Blantyre. HIV-infected patients must further be (3) ART-naïve or initiated ART <10 days prior to enrolment and (4) initiating standard first-line ART . Adult controls must further be (5) HIV-uninfected. Study exclusion criteria are (6) clinical history of CBD/CVD, (7) pregnancy, (8) critical illness or symptomatic anaemia at baseline, and (9) enrolment in an intervention study. At the analysis stage abnormal pulse wave velocity (PWV) at baseline (as defined in table 2) will be excluded for new-onset vasculopathy analysis but not for progression of vasculopathy. The same approach will be applied for baseline carotid intima-media thickness (CIMT) measurements. If the study participant becomes pregnant after recruitment, they will be withdrawn.

Justification of study inclusion and exclusion criteria is as follows: in many populations, CBD/CVD risk rises sharply from 35 years of age[33]; thus, individuals aged 35 and older will be eligible (recruitment of participants aged 35–39 will be limited to 15% of the study sample to avoid over-representation). Restricting recruitment by

**Table 1** Laboratory tests and clinical procedures in ART patients and HIV-uninfected adults

| | Study time points | | | | | | |
| --- | --- | --- | --- | --- | --- | --- | --- |
| | Baseline | 6 months | 12 months | 18 months | 24 months | 30 months | 36 months |
| **Clinical procedures** | | | | | | | |
| PWV | X | X | X | X | X | X | X |
| CIMT | X | | | | X | | |
| ABPI | X | X | X | X | X | X | X |
| Cardiac echo (participant subset) | X | | | | X | | |
| ECG (participant subset) | X | | | | X | | |
| **Cardiometabolic markers** | | | | | | | |
| Creatinine | X | X | | | X | | |
| Full blood count | X | | | | | | |
| Cholesterol (LDL, HDL, triglycerides) | X | | | | X | | |
| Serum glucose/haemoglobin A1c | X | | | | X | | |
| **HIV infection and progression** | | | | | | | |
| HIV viral load (patients with HIV) | X | X | X | | | | |
| CD4 count (patients with HIV) | X | X | X | | | | |
| HIV rapid test (controls) | X | | X | | X | | X |
| **Immune dysregulation** | | | | | | | |
| Soluble markers of systemic inflammation | X | X | X | | | | |
| Soluble markers of endothelial activation | X | X | X | | | | |
| CD8 and CD4 T cell activation and senescence (participant subset) | X | X | X | | X | | X |
| Monocyte/Macrophage activation and senescence (participant subset) | X | X | X | | X | | X |
| **Herpesviruses infection** | | | | | | | |
| CMV IgG | X | X | | | | | |
| VZV IgG | X | X | | | | | |

ABPI, Ankle Brachial Pressure Index; ART, antiretroviral therapy; CIMT, carotid intima-media thickness; CMV, cytomegalovirus; HDL, high-density lipoprotein; LDL, low-density lipoprotein; PWV, pulse wave velocity; VZV, varicella zoster virus.

age will enable this study to have greater statistical power. For clarity of aetiological inference, the study will assess the risk of new-onset vasculopathy not associated with pregnancy and thus exclude patients who are pregnant or with a history of CBD/CVD. To eliminate confounding by ART regimen, patients must initiate on standard first-line ART (>90% of ART patients in Blantyre do this). Critically ill patients are excluded primarily for ethical reasons.

## LABORATORY METHODS
### Surface immunophenotyping of peripheral blood mononuclear cells
Immunophenotyping will be used to characterise peripheral blood mononuclear cells (PBMC) isolated from blood samples of HIV-uninfected and HIV-infected ART initiators. PBMCs will be harvested by density centrifugation using Lymphoprep (Axis Shield,

UK). PBMCs ($2 \times 10^6$) will be stained with anti-CD45 PerCP CY5.5, anti-CD3 AF700, anti-CD4 BV421, anti-CD8 PE Dazzle, anti-CD38 BV605, anti-HLA-DR APC CY7, anti-CD57 APC, anti-PD1 PE CY7, anti-CTLA4 PE and anti-CD223 FITC (all from eBioscience, UK) to determine the expression of these markers on the surface of T cells. In addition, $2 \times 10^6$ PBMCs stained with anti-CD16 BV421, anti-CD14 PE, anti-HLA-DR PerCP CY5.5, anti-CD45 AF700, anti-CCR2 BV605, anti-CD11b APC, anti-CX3CR1 PE Dazzle and anti-CD38 FITC (all from eBioscience, UK) will be used for monocytes. Dead cells, CD3+ T cells and CD56+ Natural Killer cells will be excluded using LIVE/DEAD Fixable Aqua Dead Cell Stain (Thermo Fisher, UK), anti-CD3 BV503 and anti-CD56 BV503 (eBioscience, UK), respectively. Stained cells will be acquired on a BD LSRFortessa flow cytometer (Becton Dickinson,

**Table 2** Case definitions of primary and secondary endpoints for the study

| | Type | Definitions |
|---|---|---|
| Primary endpoint | | *The occurrence of new-onset vasculopathy (CIMT, a measure of atherosclerosis):* CIMT >0.9 mm or >75th percentile of age/sex references values or presence of plaque on the carotid scan. *Progression:* total change in CIMT at 24 months from baseline. |
| | Carotid intima-media thickness (CIMT) | |
| | | *Occurrence of new-onset vasculopathy (PWV, a measure of arterial stiffness):* PWV >12 (m/s). |
| | Pulse wave velocity (PWV) | *Progression:* total change in PWV at 24 months from baseline. |
| Secondary endpoint | Stroke | *Confirmed (1+2) or 3 or 4 or 5:* <br>1. Acute onset with a clinically compatible course, including unequivocal objective findings of a localising neurological deficit. <br>2. CT or MRI compatible with a diagnosis of stroke and current neurological signs and symptoms. <br>3. Stroke diagnosed as the cause of death at autopsy. <br>4. Clinical history and positive lumbar puncture compatible with subarachnoid haemorrhage. <br>5. Death certificate or death note from medical record listing stroke as the cause of death. |
| | Myocardial infarction (MI | *Confirmed: one of the following five criteria (adapted from 2007 Universal Definition of Myocardial Infarction) (1 + (2 or 3 or 6)) or 4 or 5:* <br>1. Rise and/or fall of cardiac biomarkers (preferably troponin), with at least one value above the 99th percentile of the upper reference limit. <br>2. The occurrence of a compatible clinical syndrome, including symptoms consistent with myocardial ischaemia. <br>3. ECG changes indicative of new ischaemia (new ST changes or new left bundle branch block (LBBB)), or development of pathological Q waves on the ECG. <br>4. Sudden unexpected cardiac death involving cardiac arrest before biomarkers are obtained or before a time when biomarkers appear, along with (a) new ST changes or new LBBB, or (b) evidence of fresh thrombus on coronary angiography or at autopsy. <br>5. Pathological findings of acute MI (including acute MI demonstrated as the cause of death on autopsy). <br>6. Development of (a) evolving new Q waves or (b) evolving ST elevation, preferably based on at least 2 ECGs taken during the same hospital admission. |
| | Coronary artery disease requiring drug treatment | *Confirmed (1 or 2) +3:* <br>1. Evidence of myocardial ischaemia based on either diagnostic imaging (such as a stress echocardiogram or thallium scan) or diagnostic changes on an ECG (such as during stress testing or an episode of chest pain). <br>2. Evidence of coronary artery disease based on coronary angiography or other diagnostic imaging. <br>3. Use of medication given to treat or prevent angina (eg, nitrates, beta blockers, calcium channel blockers). |
| | Peripheral vascular disease | *Confirmed (1+2) or (1+3):* <br>1. Compatible clinical signs and symptoms. <br>2. Positive results on diagnostic imaging studies (eg, Doppler ultrasound, contrast arteriography, MRI arteriography). <br>3. Ankle brachial pressure index <0.90 in non-diabetics. |
| | Vascular Immune reconstitutionsyndrome (IRIS) | A new onset vasculopathy within 6 months of starting ART |
| | All-cause death and vascular-related deaths | Death (of any or vascular cause) that occurs after recruitment into the study. |

USA), and data will be analysed using FlowJo V.10.0 software (Tree Star, San Carlos, California). For each stained sample analysed, the median fluorescence intensity for each parameter will be normalised to its respective unstained control.

### Measurement of soluble markers of immune activation using multiplex bead array

A custom-made multiplex assay will be used to assess soluble markers of monocyte activation (CD163), systemic inflammation (IL-6) and endothelial activation (intra-cellular adhesion molecule 1) in plasma, isolated from blood samples of HIV-uninfected and HIV-infected ART

initiators. Following isolation, plasma will be aliquoted and stored at −80°C until further use.

### Assessment of exposure to human CMV and VZV by ELISA

Quantitative VIDAS CMV IgG and IgM (bioMerieux, USA) and VZV glycoprotein IgG Low-Level Enzyme Immunoassay Kit (VaccZyme EIA) will be used to determine exposure to these viruses using a commercial ELISA platform. These kits will detect VZV antigen to a sensitivity and specificity of 97.8% and 96.8% and for CMV, 97.2% and 100% for IgG, and 100% and 97.4% for IgM, respectively.[34 35] Plasma samples from HIV-uninfected and HIV-infected ART initiators stored

at −80°C following collection will be used for these assessments.

### Human immunodeficiency virus

HIV infection will be diagnosed using two rapid tests in parallel (Determine HIV-1/2 (Abbott Laboratories, USA) and Uni-Gold HIV (Trinity Biotech, Ireland), and EIA rapid tests will be used as a tie-breaker. HIV-1 RNA levels in plasma will be measured using the Abbott RealTime HIV-1 assay with a lower limit of detection of 150 copies/mL (Abbott Molecular, Germany), according to the manufacturer's instructions. CD4+ T cell count measurements will be performed using BD FACSCount machine (Partec platform).

### Procedures

Carotid-femoral PWV[36] and CIMT[37] measurements will be performed in accordance with expert consensus guidelines, using a standardised study protocol on the Vicorder system (SMART Medical, UK) and Philips CX50 machine (Philips Healthcare, UK), respectively. CIMT measurements will be performed by three trained operators. The intraclass correlation coefficient will be used to assess the performance of the operators against that of a certified neurosonologist prior to study commencement.

### OUTCOMES
### Primary outcomes

The primary outcomes are the progression of surrogate markers of CBD/CVD, namely PWV and CIMT, as well as the occurrence of new-onset vasculopathy defined by threshold values outlined in table 2.

### Secondary outcomes

The secondary outcomes are the following clinical events: (1) stroke, (2) MI, (3) unstable angina, (4) PVD, (5) all-cause death/vascular death and (6) immune reconstitution inflammatory syndrome (IRIS) vasculopathy (table 2). Changes in PWV or endothelial activation at 6 months post-ART initiation will be interpreted as a subclinical vascular IRIS event. These outcomes will be assessed through active surveillance in QECH inpatient wards for admissions of study participants. To improve capture of clinical outcomes, we will conduct brief telephone interviews with study participants about CBD/CVD symptoms and hospitalisations between study visits and facilitate unsolicited participant self-report. Clinical events and deaths in study participants will be reviewed by an independent endpoint review committee (ERC), comprising clinicians experienced in endpoint review. Each event will be reviewed and adjudicated by the ERC chair and two ERC reviewers, using a standard set of diagnostic criteria (table 2 and online supplementary S1). The format of reporting will be based on modifications of the INSIGHT network clinical diagnostic criteria. Deaths will be reviewed by the ERC using the International Classification of Diseases for causes of death classificaton.[23] For death with limited clinical data, a validated verbal autopsy will be performed to ascertain the cause.[38]

### Exposures

The exposure for primary objective 1 will be HIV status. Yearly HIV rapid tests in HIV-uninfected adults will be performed to exclude those with new HIV infections (figure 2).

Potential confounding and mediating factors will be recorded in study participants. This will include demographic factors, lifestyle and behavioural factors (eg, cigarette smoking and alcohol consumption), chronic comorbidities (ie, hypertension, diabetes), and cardiometabolic, renal and haematological factors (ie, full blood count, creatinine in urine and serum, body mass index,

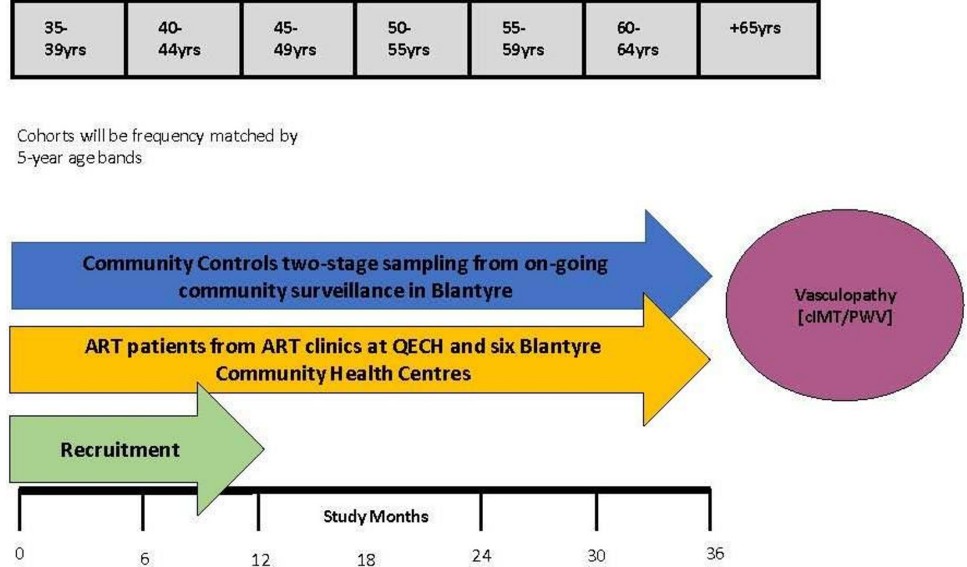

**Figure 2** Outline of study design for a 36-month cohort study. ART, antiretroviral therapy; cIMT, carotid intima-media thickness; PWV, pulse wave velocity; QECH, Queen Elizabeth Central Hospital.

waist to hip ratio, random glucose, haemoglobin A1c, and lipid profile). Blood pressure will be measured at all study visits. Although vascular IRIS (table 2) will be considered as a primary endpoint, non-vascular IRIS will be defined as a risk factor. Where feasible, we will conduct PCR tests for common causes of IRIS in blood or cerebrospinal fluid samples. Adherence to ART and change of ART regimen will be assessed at all study visits through extraction of data from 'ART master cards'; this is a government-supported monitoring tool used by all patients on ART in Malawi.

For objectives 2a–2c, markers of herpes and HIV viral antigenaemia and immune inflammation will be measured according to the outline in table 1. For primary objective 2a, reactivated latent herpes viral infections will be assessed by quantification of VZV and CMV antibodies. We will estimate the risk of atherosclerosis and arterial stiffening associated with current herpesviruses reactivation at baseline and sustained reactivation (ie, those that continue to have a high titre from measurement at baseline to 6 months after ART initiation). Hyperactivation of B cells may result in an expansion of polyclonal antibodies and thus an overestimation of virus-specific antibody titres. To address this issue and make appropriate adjustments for hypergammaglobulinaemia, we will (1) measure more than one herpesviruses and (2) measure total IgG.

For primary objective 2b, markers of immune activation, inflammation and endothelial activation will be measured (figure 1 and table 1). Quantitative cell surface immunophenotyping will be performed for CD4+ and CD8+ T cell activation (eg, Human Leukocyte Antigen – DR isotype [HLA-DR]) and senescence (eg, CD57) in a subset of participants. In all study participants, at baseline and at 6 and 12 months, we will measure soluble markers associated with systemic inflammation and endothelial activation.

For primary objective 2c, incomplete viral response and viral rebound of HIV will be measured by quantitative PCR in patients on ART.[39] HIV viral load will be measured in patients on ART at 0, 6 and 12 months.

### Data collection between May 2017 and August 2020
The two-stage screening will be conducted to find and recruit potential study participants. A trained field-worker will first screen to assess eligibility for criteria 1–3 in pre-ART counselling sessions and in individuals from randomly selected households in the community. Eligible participants will then be referred to the QECH to complete screening for criteria 4–9, and if eligible consented to participate in the study. At study visits, a tablet-based, standardised Open Data Kit case report form will be administered in one-on-one interviews by a study nurse to capture demographic and clinical data. Study data will be collected as outlined in table 1. Daily upload of electronic data will occur with oversight from the data manager at MLW. We will collect up to 30 mL of whole blood. Urine albumin to Creatinine ratio (ACR) dipstick test will be used to test for creatinine, proteinuria and glucosuria. In

a subset of participants, an ECG supported by a digital platform and echocardiogram will be performed at baseline and at 6 and 24 months, as well as in any participant experiencing a clinical event suggestive of a cardiac aetiology. To facilitate the retention and clinical referrals of participants, contact will be made every 3 months to assess the occurrence of clinical events. Participants who miss a scheduled study visit will be contacted by phone and/or visited at home to assess their willingness to maintain their participation and to record intervening clinical events. Recording and definitions of other clinical events, including HIV-associated events, will be evaluated by the ERC chair. SMS (short message service) messages will be used for appointment reminders. A technical appendix, statistical code and study data sets will be made available from a data repository, after publication of our work.

### Sample size and statistical analysis
The required sample size for the study's primary objectives is 800 HIV-infected patients and 190 HIV-uninfected adults using standard, normal distribution approximation sample size formulas for comparing proportions in two groups of unequal size and based on the following assumptions: (1) of HIV-infected study participants, 18.4% have abnormal PWV at baseline. We will exclude these participants from analysis. The 18.4% figure is informed by our ongoing studies of vasculopathy in HIV-infected patients, where this is the percentage of participants aged ≥35 years who have a PWV (>12 m/s). (2) Of both HIV-infected patients and HIV-uninfected adults, 20% will be lost to follow-up, including by death and HIV seroconversion.[40 41] (3) The minimum relative risk (RR) of interest is 2 for objective 1 and 1.8 for objective 2. (4) The 24-month cumulative risk of clinically significant vasculopathy over study follow-up is 18.4% in the HIV-positive group. This is based on the same study data cited in (1). (5) For objectives 2a–2c, the exposure prevalence for each risk factor is 20%. (6) Statistical tests will have 80% power based on a two-sided test with α=0.05. Testing of hypotheses for the secondary outcome will be exploratory. However, we estimate 26 strokes (4 mimics), an unknown number of MIs and 80 deaths occurring during the study.[7 42] Taken together, assumptions (3), (4) and (5) mean that, for 80% power, we assume a 24-month cumulative vasculopathy risk of 9.2% in HIV-negative participants, 18.4% in all HIV-infected participants, 15.9% in HIV-infected participants not exposed to the risk factors from objectives 2a–2c and 28.6% in the HIV-infected participants exposed to these risk factors.

The reporting of this study will be prepared in accordance with the Strengthening the Reporting of Observational Studies in Epidemiology guidelines.[43] Summary and descriptive statistics will be tabulated for all primary and secondary outcome variables, as well as for exposure variables and potential confounding or mediating factors. Time plots for all outcome variables will be inspected. Quantitative data analysis will be conducted to assess the primary outcomes.

There will be three analysis time points: (1) after recruitment has finished and baseline data are available for all participants (baseline analysis); (2) once every participant has completed 6 months in the study (6-month analysis); and (3) at 36 months, when each participant has completed 24 months in the study (final analysis).

The baseline analysis will largely consist of descriptive statistics on participant characteristics and data recorded at baseline. Simple regression models will also be used to investigate relationships between exposure and outcome variables measured at baseline. Unadjusted analyses will consist of paired t-tests or Wilcoxon signed-rank tests (depending on whether the data are normally distributed or not) for continuously measured variables, and $\chi^2$ or Fisher's exact tests (depending on contingency table cell counts) for binary and categorical variables. Adjusted analyses will be conducted using generalised linear models (GLMs).

The 6-month analysis will be limited in scope and serves two purposes: (1) characterise new-onset vasculopathy in HIV-infected participants who have initiated ART treatment at baseline (vascular IRIS) and (2) define vasculopathy outcomes for the final analysis. The main analysis of the study data will be conducted at the final analysis time point.

For objective 1 we will develop three regression models. Two GLMs will be developed to compare mean progression of arterial damage from baseline in HIV-infected ART patients and HIV-uninfected adults. These models will regress change from baseline PWV and CIMT, respectively, on HIV status. We will develop a third model to estimate the RR and population-attributable fraction of new-onset arterial damage in HIV-infected patients compared with HIV-uninfected adults.

For objective 2a, a set of GLMs will be developed to compare mean progression of vasculopathy in HIV-infected ART patients with and without reactivated latent herpes viral infection. These models will regress PWV and CIMT, respectively, on, on two log-transformed variables for antibody titres of CMV and VZV.

For objective 2b, we will again fit a set of GLMs, with change from baseline in PWV as response variable, this time to investigate if, in HIV-infected ART patients, there is an association between progression of vasculopathy and immune activation and inflammation biomarkers (IL-6, Intracellular adehesion molecule [ICAM], CD163). Specifically, for each marker, we will regress PWV on marker quantiles. After having built models for each marker, we will then develop comprehensive multiple regression models for PWV and CIMT with multiple independent markers as predictor variables.

For objective 2c, we will proceed as for objective 2a, but comparing HIV-infected ART patients with incomplete virological suppression or virological resurgence of HIV with those with suppressed HIV plasma viral load.

In addition to these analyses, given the repeated measurements for PWV, immune activation and inflammation markers, we will extend the GLMs for PWV to linear mixed models taking full account of the longitudinal nature of the data. Mixed models will also handle deviations from the visit schedule in a principled fashion and use all available data for dropouts. If a log link function is required for PWV in the GLMs, we will fit marginalised models using Generrlaised Estimating Equations [GEE] instead of Linear Mixed Models [LMMs].

For the secondary study objectives, we will use univariate methods to assess the frequency of clinical events within exposure strata. If there are sufficient numbers of clinical events, we will develop Poisson or negative binomial regression models (depending on model fit) for each clinical event type to compare exposure-defined participants.

We will also use time-to-event models, specifically Cox proportional hazard models, to investigate associations between all-cause mortality and exposures.

As part of exploratory analyses, we will aim to identify risk groups that are potentially incompletely captured with the measured exposure variables. We will perform unsupervised group-based multitrajectory modelling of multivariate longitudinal patient trajectories to confirm any associations we have found using more traditional approaches.[44]

All efforts will be made to collect complete data on all study participants. However, there will inevitably be missing data due to dropouts and a variety of other reasons. All primary analyses will be performed using multiple imputation. For sensitivity analyses, we will use all available cases, direct likelihood and fully Bayesian models, and for GEE models weighted GEE. If the number of missingness patterns is sufficiently small, we will also use pattern mixture models which can be used under the general missing-not-at-random setting but make additional identification assumptions.

## PATIENT PUBLIC INVOLVEMENT

The global burden of HIV-associated CBD and CVD has tripled over the last two decades with the greatest impact in SSA. CBD and CVD are a priority for patients in Malawi as HIV infection is endemic and the population are living for longer. Knowledge of this informed our research question with the aim of understanding the mechanisms and thus direct targeted novel therapies to reduce this burden. Patients will be involved in the recruitment of the study, but not in the design. Patients and their advisors will be thanked for contributing to the study.

## ETHICS AND DISSEMINATION

Written informed consent will be obtained from all study participants, either written or witnessed verbal consent with thumbprint if the participant is non-literate. Study data will be maintained in an encrypted and password-protected database to which only study staff will have access. Study participants who develop a clinical event will be managed, using the hospital guidelines, by our study clinician alongside the hospital doctor. Clinical data

will be anonymised using unique identifying code. Study data will be kept for 10 years and then destroyed with a record, as recommended by good clinical practice guidelines. Results of the main trial and each of the secondary endpoints will be submitted for publication in a peer-reviewed journal.

## DISCUSSION

African regions continue to bear the brunt of HIV infection. In 2013, an estimated 8.5 million adults were receiving ART.[45] As the landscape evolves, this population will live longer with stable HIV infection but likely remain at an increased risk of CBD/CVD compared with HIV-uninfected individuals of similar age and sex. This study will be the first to determine the extent to which HIV reactivation of herpesvirus infection and inflammation contribute to CBD/CVD risk in an adult African population starting ART. The results of this work could potentially open avenues for novel anti-inflammatory and antiviral interventions for the primary prevention of CBD/CVD in HIV populations in Africa.

**Author affiliations**
[1]Liverpool School of Tropical Medicine, Liverpool, UK
[2]Malawi Liverpool Wellcome Trust Clinical Research Programme, Blantyre, Malawi
[3]Department of Medicine, University of Cape Town, Cape Town, South Africa
[4]School of Medicine, University College Dublin, Dublin, Ireland
[5]Institute of Infection and Global Health, University of Liverpool, Liverpool, UK
[6]Institute of Infection and Global Health, University College London, London, UK
[7]Kirby Institute, University of New South Wales, Sydney, New South Wales, Australia
[8]Department of Medicine, University of Malawi College of Medicine, Blantyre, Malawi
[9]Dignitas International, Zomba, Malawi
[10]Division of Infection and Immunity, University College London, London, UK
[11]Institute of Infection and Global Health, University of Liverpool, Liverpool, UK
[12]Department of Brain Repair and Rehabilitation, Institute of Neurology, University College London, London, UK

**Correction notice** This article has been corrected since it was published.

**Acknowledgements** The authors would like to thank Agbor Ako and Maria Davy from Research and Development, GlaxoSmithKline and the NCD Africa Open Lab of GlaxoSmithKline review committee for providing valuable advice for this protocol. The authors would like to thank BA, MC, LH,and T-HC for their contribution to the endpoint review committee, RD and EJ for radiology training and quality control, EZS for providing an ECG platform and for his cardiology review, VK for input with the echocardiogram protocol, and TS, JM, KM and MN for their input in the advanced drafts of the manuscript. The authors also extend their gratitude to the INSIGHT network for sharing their clinical endpoint criteria.

**Collaborators** Brian Angus (Oxford Centre for Clinical Tropical Medicine, University of Oxford), Myles Connor (University of Edinburgh), Reena Dwivedi (Greater Manchester Neurosciences Centre, Salford Royal Foundation Trust), Lewis Haddow (Institute for Global Health, University College London), Terttu Heikinheimo-Connell (Hyvinkää Hospital, Department of Neurology, University of Helsinki), Elizabeth Joekes (Liverpool School of Tropical Medicine), Vanessa Kandoole (Department of Medicine, University of Malawi College of Medicine, Blantyre), Moffat Nyrienda (MRC Research Unit, Uganda), Kennedy Malisita (Department of Medicine, Queen Elizabeth Central Hospital), Jane Mallewa (Department of Medicine, University of Malawi College of Medicine, Blantyre), Elsayed Z Soliman (School of Medicine, Wake Forest School of Medicine), Tom Solomon (Institute of Infection and Global Health, University of Liverpool).

**Contributors** LB and IP developed the first draft. HM, NN, KJ, CK, LA, JKT, SP, MH, JVO and RSH had major input for the revision of the second draft. JH is the project manager for RHICCA with oversight from LB, IP and HM. MH contributed to the statistical methods. LB, JKT and JVO contributed to the clinical training. SP chaired the endpoint review committee.

**Funding** LB is supported by an NIHR Clinical Lecturer Fellowship. SP is supported by an MRC (UK) core funding (MC_UU_12023/23). Funding for this study was provided by the GlaxoSmithKline Africa Non-Communicable Disease Open Lab Grant (Project Number: 7964).

**Competing interests** SP has academic grants from Sysmex, Gilead Sciences and ViiV Healthcare.

**Patient consent for publication** Not required.

**Ethics approval** This protocol was approved by the ethics committees at the University of Malawi College of Medicine (Protocol P02/16/1874) and the Liverpool School of Tropical Medicine (Protocol 16–014).

**Provenance and peer review** Not commissioned; externally peer reviewed.

**ORCID iDs**
Joseph Kamtchum Tatuene http://orcid.org/0000-0002-1041-4202
Laura A Benjamin http://orcid.org/0000-0002-9685-1664

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
