## [Reviewer comments · BMJ Open]

ARTICLE DETAILS

TITLE (PROVISIONAL)	Protocol for a longitudinal cohort study to evaluate the reactivation of herpesviruses and inflammation as cardiovascular and cerebrovascular risk factors in antiretroviral therapy initiators, in an African HIV-infected population (RHICCA)
AUTHORS	Peterson, Ingrid; Ntsui, Ntobeko; Jambo, Kondwani; Kelly, Christine; Huwa, Jacqueline; Afran, Louise; Tatuene, Joseph; Pett, Sarah; Henrion, Marc; Van Oosterhout, Joep; Heyderman, Robert; Mwandumba, Henry; Benjamin, Laura

VERSION 1 - REVIEW

REVIEWER	Janet Raboud University of Toronto, Canada
REVIEW RETURNED	07-Aug-2018

GENERAL COMMENTS	This is an important and well-designed study to examine the associations of HIV infection, immune activation and reactivation of latent herpesvirus infection with risk of cerebrovascular and cardiovascular disease. I have the following comments and suggestions with regards to the statistical analysis plans. 1. For the sample size calculations:a. Please clarify if the expected rate of clinically significant vasculopathy of 18.4% is among the HIV positive group or the control group.b. It is noted that 18.4% of study participants have abnormal PWV at baseline. Will these patients be excluded from the analyses?2. Please clarify if participants will be withdrawn from the study if they become pregnant during the study period.3. It is my understanding that the investigators are using the term "interim analysis" to refer to an analysis to be conducted of data up to an interim follow-up visit, rather than of the complete data. Since they have not specified otherwise, I assume this analysis will take place after all participants have reached the interim time point. It may be less confusing for readers if the investigators refer to these analyses as "6-month analyses" or "12-month analyses", for example, to distinguish them from interim analyses conducted in a clinical trial context, which would be accompanied by specific stopping rules, etc.4. Will missing data be imputed?5. The investigators may wish to consider the use of pattern mixture models to assess the affect of differential dropout by HIV status, reactivation of herpes viral infection and HIV viral
---

	antigenemia on the estimated effects of these characteristics on risk of cerebrovascular and cardiovascular disease. 6. Will the exclusion of study participants who die from analyses depend on whether study participants die from CBD or CVD? 7. In the study design section, it is indicated that the study schedule is described in "Figure 2". Please correct this to "Table 2". 8. It is indicated that enrollment for the study started in May 2017 and will continue until August 2020. Will this be sufficient time to enroll 990 participants and follow them for 36 months each?
--	--

REVIEWER	ISAAC SSINABULYA Makerere University College of Health Science, Uganda
REVIEW RETURNED	15-Aug-2018

GENERAL COMMENTS	Great initiative authors; Just a small comment: On page 15, you state that in a subset of participants some tests will be done but you make no mention how these participants will be selected. Same for page 5, the small number that will have an MRI.
---

VERSION 1 – AUTHOR RESPONSE

Reviewers' Comments to Author:

Reviewer: 1

Reviewer Name: Janet Raboud

Institution and Country: University of Toronto, Canada

Please state any competing interests or state 'None declared': None declared

This is an important and well-designed study to examine the associations of HIV infection, immune activation and reactivation of latent herpesvirus infection with risk of cerebrovascular and cardiovascular disease.

RESPONSE: We thank the reviewer for these comments.

I have the following comments and suggestions with regards to the statistical analysis plans.

1. For the sample size calculations:

a. Please clarify if the expected rate of clinically significant vasculopathy of 18.4% is among the HIV positive group or the control group.

RESPONSE: This is the expected rate in the HIV positive participants. We have clarified this in the sample size paragraph.

b. It is noted that 18.4% of study participants have abnormal PWV at baseline. Will these patients be excluded from the analyses?

RESPONSE: We will be conducting two types of analysis. The first will examine the progression of surrogate markers over time from whatever baseline value - participants with abnormal PWV at baseline will not be excluded from this analysis. The second will examine the development of new-onset vasculopathy - participants with abnormal PWV at baseline will be excluded from this analysis. We have clarified this point in the manuscript.

2. Please clarify if participants will be withdrawn from the study if they become pregnant during the study period.

RESPONSE: We confirm that if the study participant becomes pregnant after recruitment, they will be withdrawn. Given the age structure of our population, we do not think that this will have a substantial impact on our lost to follow-up numbers. We have clarified this in the manuscript.

3. It is my understanding that the investigators are using the term “interim analysis” to refer to an analysis to be conducted of data up to an interim follow-up visit, rather than of the complete data. Since they have not specified otherwise, I assume this analysis will take place after all participants have reached the interim time point. It may be less confusing for readers if the investigators refer to these analyses as “6-month analyses” or “12-month analyses”, for example, to distinguish them from interim analyses conducted in a clinical trial context, which would be accompanied by specific stopping rules, etc.

RESPONSE: We thank the reviewer for pointing out this somewhat confusing notation and we have adopted instead of the notation suggested by the reviewer. Further we have clarified what we mean with each analysis timepoint. We also explicitly state that there will be descriptive analyses performed once baseline data is available on all participants. As part of the development of the statistical analysis plan, we have also limited the scope of the 6-month analysis considerably and this is now reflected in the updated analysis section of the manuscript.

4. Will missing data be imputed?

RESPONSE: This is included in the statistical analysis plan we have developed but was left out of the brief summary for this protocol paper. We expect a sizeable number of drop-outs and will potentially also have other missing values in the data. For the primary analyses we will use multiple imputation (valid under missing-at-random) as this can be used across all analyses we are planning (unadjusted group comparisons, GLMs, LMMs and GEEs) but plan to perform all-available-cases analyses (requiring the stronger assumption of missing-completely-at-random), direct likelihood and fully

Bayesian analyses where applicable as sensitivity analyses. Mixed models will anyway be direct likelihood models and we can also use weighted GEE as part of the sensitivity analyses for the GEE models.

We have added a short paragraph to the statistical analysis section of the protocol paper.

5. The investigators may wish to consider the use of pattern mixture models to assess the affect of differential dropout by HIV status, reactivation of herpes viral infection and HIV viral antigenemia on the estimated effects of these characteristics on risk of cerebrovascular and cardiovascular disease.

RESPONSE: We thank the reviewer for this suggestion. While we will focus on the selection modelling framework, we do hope to use pattern mixture models to gain further insights. However, this will only be feasible if there are sufficiently few distinct missingness patterns. Most of the missing data should be due to drop-out, which should limit the number of missing data patterns. This is also mentioned in the paragraph on missing data we have added to the manuscript.

6. Will the exclusion of study participants who die from analyses depend on whether study participants die from CBD or CVD?

RESPONSE: We will conduct several types of analysis of these complex longitudinal data. We will use GEE models to examine progression of surrogate markers over time. These models can accommodate the occurrence of missing data at a given time point. For example, if a participant died after the 9-month time point, all their data up to 9 months would be included in the model, regardless of their cause of death.

7. In the study design section, it is indicated that the study schedule is described in "Figure 2". Please correct this to "Table 2".

RESPONSE: This has been amended, many thanks for highlighting.

8. It is indicated that enrollment for the study started in May 2017 and will continue until August 2020. Will this be sufficient time to enroll 990 participants and follow them for 36 months each?

RESPONSE: The minimum follow-up requirement is 24 months which all participant will achieve.

Reviewer: 2

Reviewer Name: ISAAC SSINABULYA

Institution and Country: Makerere University College of Health Science, Uganda

Please state any competing interests or state 'None declared': None

Great initiative authors; Just a small comment:

RESPONSE: We thank the reviewer for his kind comment.

On page 15, you state that in a subset of participants some tests will be done but you make no mention how these participants will be selected. Same for page 5, the small number that will have an MRI.

RESPONSE: Thank you for highlighting this oversight. We have now expanded on this in the manuscript.

We expect a subset of patients to have clinical events and thus be 'symptomatic'. These patients will be investigated to establish a diagnosis and validated by the ERC members (supplement Table 1). Patients with focal neurological clinical event or those with cognitive clinical symptoms will have an MRI head.

VERSION 2 – REVIEW

REVIEWER	Janet Raboud University of Toronto, Canada
REVIEW RETURNED	01-Mar-2019

GENERAL COMMENTS	The authors have answered most of my questions from my first review of this paper. I have just a few further questions on the duration of follow-up and the calculation of the sample size. 1. The authors have clarified that all participants will achieve 24 months of follow-up. In the abstract and 'Study Design' section, the study is described as a '36-month prospective cohort study'. If 36 refers to the duration of time that the study will be under way, rather than the planned length of follow-up for all participants, then it would be clearer to describe the study as a '24-month prospective cohort study'. If the planned duration of follow-up for each individual is 36 months but not all individuals will achieve that due to time constraints, then the sample size calculation needs to be modified (see below). 2. In section (a) of the sample size section, the authors have clarified that 18.4% of HIV+ individuals have abnormal PWV at baseline, and that these individuals will be excluded from the analysis of new vasculature. Does the second sentence in (a) mean that 18.4% of HIV- individuals will be also excluded from this analysis, or does this sentence refer to expected rates of events during follow-up? In point (d) of the assumptions, is the cumulative risk of clinically significant vasculature over study follow-up of 18.4% the rate for HIV+ or HIV- individuals? If the planned duration of follow-up is 36 months but some participants will only be able to achieve 24 months of follow-up due to time constraints, the length of follow-up should be
---

	accounted for in the sample size calculations. A sample size calculation to compare proportions of individuals between groups would be reasonable if most participants are expected to be followed for the same duration. Please clarify if the sample size calculations assume that individuals who are lost to follow-up due to death or HIV seroconversion will be excluded from analyses, or included for the portion of the study for which they were observed.
--	--

VERSION 2 – AUTHOR RESPONSE

Reviewer: 1

Reviewer Name: Janet Raboud

Institution and Country: University of Toronto, Canada

Please state any competing interests or state 'None declared': None declared

Please leave your comments for the authors below

The authors have answered most of my questions from my first review of this paper. I have just a few further questions on the duration of follow-up and the calculation of the sample size.

1. The authors have clarified that all participants will achieve 24 months of follow-up. In the abstract and 'Study Design' section, the study is described as a '36-month prospective cohort study'. If 36 refers to the duration of time that the study will be under way, rather than the planned length of follow-up for all participants, then it would be clearer to describe the study as a '24-month prospective cohort study'. If the planned duration of follow-up for each individual is 36 months but not all individuals will achieve that due to time constraints, then the sample size calculation needs to be modified (see below).

#Response to sample size point raised here is given in 2. below.

2. In section (a) of the sample size section, the authors have clarified that 18.4% of HIV+ individuals have abnormal PWV at baseline, and that these individuals will be excluded from the analysis of new vascularity. Does the second sentence in (a) mean that 18.4% of HIV- individuals will be also excluded from this analysis, or does this sentence refer to expected rates of events during follow-up?

These 18.4% of HIV+ individuals will be excluded from analysis. 18.4% is also the proportion of participants (over study follow-up) that are expected to develop clinically significant vasculopathy and we state this in point (d).

In point (d) of the assumptions, is the cumulative risk of clinically significant vascularity over study follow-up of 18.4% the rate for HIV+ or HIV- individuals?

#This is the 24 months cumulative risk in the HIV+ participant population. For Objective 1, with a RR of 2, this means we assumed 9.2% in the HIV- control group. For Objective 2, considering that 20% of the HIV+ population would be exposed to the risk factor, this translates to 15.9% in the non-exposed versus 28.6% in the exposed.

If the planned duration of follow-up is 36 months but some participants will only be able to achieve 24 months of follow-up due to time constraints, the length of follow-up should be accounted for in the sample size calculations. A sample size calculation to compare proportions of individuals between groups would be reasonable if most participants are expected to be followed for the same duration.

#We took a conservative approach during sample size calculation and assumed all participants to be followed up only for 24 months, comparing proportions of new onset vasculopathy over that time period rather than rates of vasculopathy events.

Please clarify if the sample size calculations assume that individuals who are lost to follow-up due to death or HIV seroconversion will be excluded from analyses, or included for the portion of the study for which they were observed.

#Here we also took a conservative approach and assumed all individuals lost to follow-up are completely excluded from the analysis.

#We thank the reviewer for their questions. We have incorporated the above changes into the manuscript as well.